# Suppressing MTERF3 inhibits proliferation of human hepatocellular carcinoma via ROS-mediated p38 MAPK activation

Zhihai Zheng[1,7], Youjuan Zhao[2,7], Hongjia Yu[2,7], Tingting Wang[3], Jinhai Li[4], Liang Xu[2], Chunming Ding[2], Lan He [5✉], Lijun Wu [6✉] & Zhixiong Dong [2✉]

Mitochondrial transcription termination factor 3 (MTERF3) negatively regulates mitochondrial DNA transcription. However, its role in hepatocellular carcinoma (HCC) progression remains elusive. Here, we investigate the expression and function of MTERF3 in HCC. MTERF3 is overexpressed in HCC tumor tissues and higher expression of MTERF3 positively correlates with poor overall survival of HCC patients. Knockdown of MTERF3 induces mitochondrial dysfunction, S-G2/M cell cycle arrest and apoptosis, resulting in cell proliferation inhibition. In contrast, overexpression of MTERF3 promotes cell cycle progression and cell proliferation. Mechanistically, mitochondrial dysfunction induced by MTERF3 knockdown promotes ROS accumulation, activating p38 MAPK signaling pathway to suppress HCC cell proliferation. In conclusion, ROS accumulation induced by MTERF3 knockdown inhibits HCC cell proliferation via p38 MAPK signaling pathway suggesting a promising target in HCC patients.

[1] Department of Colorectal Surgery, The First Affiliated Hospital of Wenzhou Medical University, 2 Fuxue Lane, Wenzhou, Zhejiang 325000, China.
[2] Zhejiang Provincial Key Laboratory of Medical Genetics, Key Laboratory of Laboratory Medicine, Ministry of Education, China, School of Laboratory Medicine and Life Science, Wenzhou Medical University, Wenzhou 325000 Zhejiang, China. [3] Department of Gastroenterology, The First Affiliated Hospital of Wenzhou Medical University, Wenzhou, China. [4] Department of Liver and Gall Surgery, The Third Affiliated Hospital of Wenzhou Medical University, Wenzhou 325200 Zhejiang, China. [5] School of Biomedical Science, Hunan University, Changsha, Hunan 410013, PR China. [6] Department of Hepatobiliary Surgery, The First Affiliated Hospital of Wenzhou Medical University, Wenzhou 325000 Zhejiang, China. [7]These authors contributed equally: Zhihai Zheng, Youjuan Zhao, Hongjia Yu. ✉email: helan2019@hnu.edu.cn; wumolijun@163.com; dongzx882@163.com

Hepatocellular carcinoma (HCC) is the most common histologic type of primary liver cancer, and ranks the third leading cause of cancer-related mortality worldwide[1]. Despite great advantages in diagnosis and therapeutic approach, the 5-year survival rate of HCC patients remains unsatisfied due to the high rate of postsurgical recurrence and metastasis[2,3]. Meanwhile, the molecular events driving HCC progression are still not fully illuminated. Therefore, it is urgent to explore the critical molecular events of HCC and search for novel biomarkers that can be used for prognostic prediction and therapy of HCC.

The liver is an organ with abundant mitochondria, serving as a metabolic hub for the tricarboxylic acid cycle, β-oxidation, respiratory activity, and adenosine triphosphate synthesis[4]. Therefore, mitochondria are vital for the normal physiological function of the liver. Meanwhile, mitochondria also closely control the development of liver fibrosis and cirrhosis via regulating cell death in chronic liver disease[5]. For example, it has been observed increased mitochondrial mass and biogenesis in liver tissues from patients with Non-alcoholic fatty liver disease and nonalcoholic steatosis heptitis[6]. Therefore, there is growing evidence to sustain that mitochondrial dysfunction is closely related to the pathogenesis of HCC[7]. Abnormal expression of several mitochondrial dynamics regulators, including Mitofusin 1, Mitofusin 2, optic atrophy gene 1, and dynamin-related protein 1, etc., were observed in HCC patients, and all of which participate in regulating HCC progression[8–10].

Gene copy number aberrations are frequently observed in solid tumor and wildly confirmed to contribute to carcinogenesis by a copy number-induced alterations in gene expression[11]. Previous study found that the long arm of chromosome 8 is frequently amplified in a large proportion of HCC[12]. We sought to explore the role of mitochondrial dysfunction in HCC tumorigenesis, and found that the human mitochondrial transcription termination factor 3 (MTERF3) localizes at 8q22.1 and is frequently amplified in HCC. MTERF3, the most conserved member of mitochondrial transcription termination factor family, has been reported to negatively regulate the transcription of mitochondrial DNA (mtDNA)[13]. It has been reported that MTERF3 is upregulated in several cancers including breast cancer, brain glioma and colorectal cancer, and its high expression is correlated with poor outcomes in patients with those types of cancer[14–16]. However, the role of MTERF3 in HCC tumorigenesis remains elusive.

Here, we found that MTERF3 is frequently elevated in HCC, and its higher expression is positively correlated with poorer prognosis in HCC patients. Suppression of MTERF3 induced mitochondrial dysfunction and reactive oxygen species (ROS) accumulation, which caused cell proliferation inhibition, cell cycle arrest and apoptosis via activation of p38 mitogen-activated protein kinase (MAPK) pathway.

## Results

### Elevated expression of MTERF3 positively correlates with poor prognosis of HCC patients.
Previous study showed that the long arm of chromosome 8 is frequently amplified in HCC[12]. We analyzed several mitochondrial related genes and found that MTERF3 localizes at 8q22.1. TCGA data showed that MTERF3 is a frequently amplified gene in several types of cancers, and the amplified frequency of MTERF3 is about 7.53% in HCC (Supplementary Fig. 1a). To assess the role of MTERF3 in HCC progression, we analyzed the expression of MTERF3 in TCGA cohort and CPTAC cohort using the online database (http://ualcan.path.uab.edu/index.html), and the results showed that both mRNA and protein of MTERF3 were significantly increased in HCC samples as compared with adjacent normal liver tissue

samples (P < 0.001, Fig. 1a, b). Consistently, we enrolled 50 patients with primary HCC, and the expression of MTERF3 mRNA in tumor tissues and corresponding adjacent normal tissues were detected using quantitative real-time PCR (qRT-PCR). The results showed that MTERF3 is markedly increased in tumor tissues (P < 0.05, Fig. 1c, d), with 36 cases showing upregulation of MTERF3 in tumor tissue (Fig. 1c). 12 cases with bigger sample were further used to detect the expression of MTERF3 protein, and also displayed a significantly elevated of MTERF3 in most of tumor samples (Fig. 1e–g). We further analyzed the relationship between MTERF3 expression and different clinicopathological parameters, and found that, consistent with TCGA data, MTERF3 expression is positively correlated with TNM stage and displayed higher expression in tumor tissues from patients with advanced HCC (P = 0.046, Table 1 and Supplementary Fig. 1b).

We further evaluated the correlation between MTERF3 expression and clinical follow-up by Kaplan-Meier analysis and log-rank test (https://kmplot.com/analysis/), and the results showed that evaluated expression of MTERF3 mRNA was associated with unfavorable overall survival (P = 0.0007), relapse-free survival (P = 0.046) and progression-free survival (P = 0.045) in the TCGA HCC cohort (Fig. 1h and Supplementary Fig. 1c, d). Collectively, these data suggest that MTERF3 is frequently upregulated and highly expressed of MTERF3 predicts a worse outcome of HCC patients.

### MTERF3 positively regulates cell proliferation in HCC cells.
Given the close relationship between MTERF3 with prognosis of HCC patients, we sought to determine the functions of MTERF3 in HCC cells. We first examined the expression of MTERF3 in HCC cells. The results showed that the protein and mRNA of MTERF3 were commonly increased in all tested HCC cells as compared to non-malignant LO2 cells (Fig. 2a, b).

MTERF3 was knockdown using a specific siRNA targeting MTERF3 (siMTERF3), and western blot results confirmed that siMTERF3 transfection could effectively suppress MTERF3 expression in both HCC-97H and LM3 cells (Fig. 2c). The results of cell count and colony formation revealed that MTERF3 knockdown markedly inhibited the proliferation and colony formation of HCC-97H and LM3 cells (Fig. 2d–g). We also established a MTERF3 knockdown cell line using a lentivirus system. Western blot results showed that lentivirus containing shMTERF3 infection effectively silenced MTERF3 expression in HCC-97H and LM3 cells (Supplementary Fig. 2a, b), and MTERF3 knockdown cells displayed a weaker proliferative capacity as compared with controls (Supplementary Fig. 2c, d). To ascertain whether MTERF3 is specific and crucial for HCC cells proliferation, another siRNA (siMTERF3-1) was synthesized and used to explore its effect on cells proliferation. The results showed that siMTERF3-1 also effectively inhibited the expression of endogenous MTERF3 and suppressed cell proliferation in both HCC-97H and LM3 cells (Supplementary Fig. 2e–h). Meanwhile, re-expression of MTERF3 could effectively restore the proliferation capacity suppressed by transfection with a specific siRNA targeting 3'-untranslated region of MTERF3 mRNA (Fig. 2h and Supplementary Fig. 2i). Conversely, a stably MTERF3 overexpressed Huh7 cells displayed a strikingly advantage in cell proliferation as compared with controls (Supplementary Fig. 3).

To further assess the effect of MTERF3 on HCC tumor growth in vivo, HCC-97H-shMTERF3 cells and corresponding control cells were injected subcutaneously into the right flank of nude mice. The results showed that MTERF3 knockdown significantly inhibited tumor growth in vivo (Fig. 2i–l). Altogether, these results indicate that MTERF3 positively regulates the proliferation of HCC cells in vitro and in vivo.

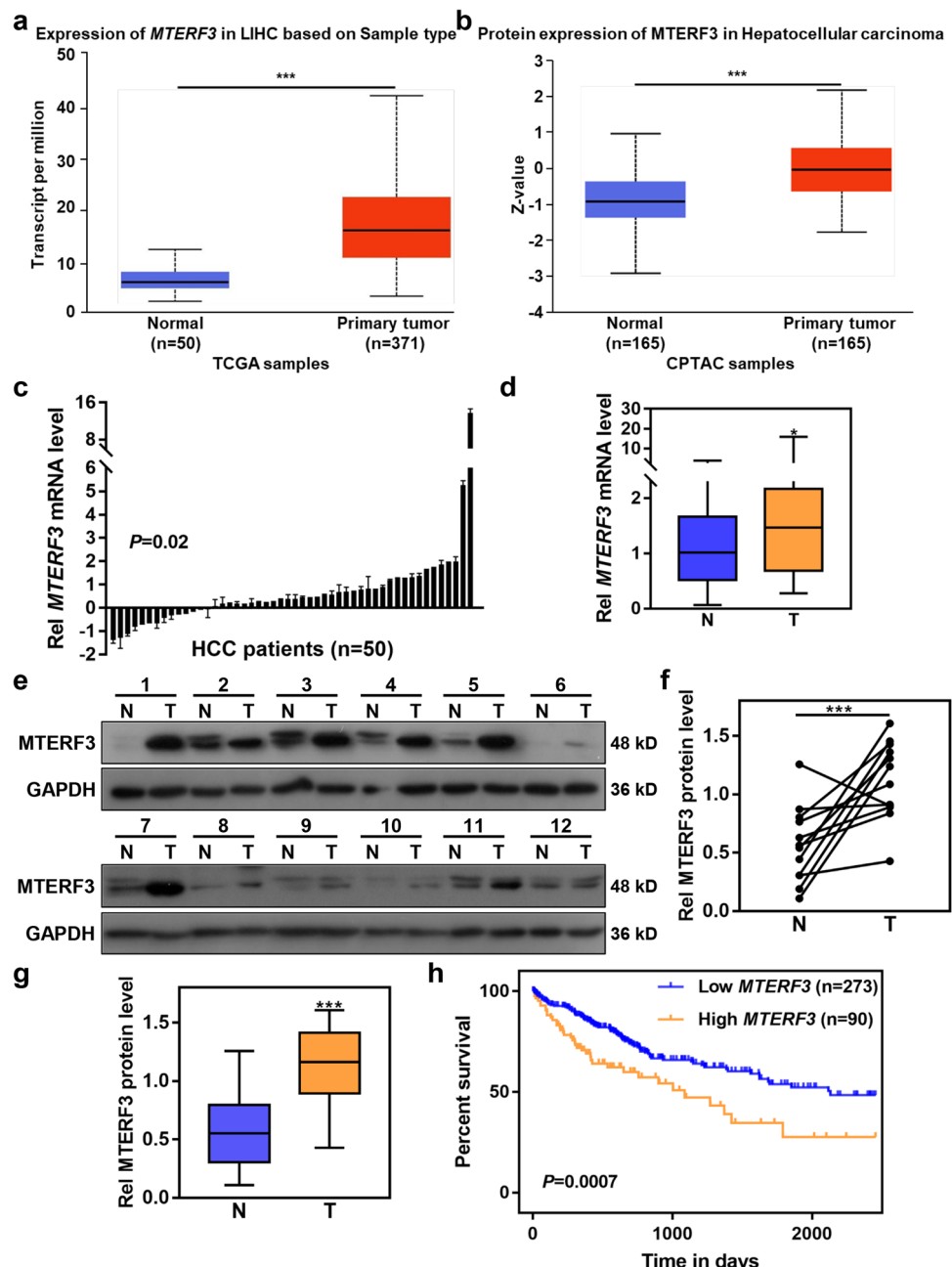

**Fig. 1 MTERF3 is frequently upregulated in HCC and displays a negative correlation with prognosis of HCC patients.** The expression of MTERF3 was analyzed using TCGA (**a**) and CPTAC (**b**) cohorts. **c** qRT-PCR was used to detect the expression of *MTERF3* mRNA in tumor samples (T, $n = 50$) and paired adjacent non-cancerous tissues (N, $n = 50$) from patients with HCC. The relative *MTERF3* expression of each patient (T/N) was shown. **d** The expression of *MTERF3* mRNA in 50 cases of HCC patients was calculated. **e** Western blot was used to examine MTERF3 expression in tumor samples (T) and paired adjacent non-cancerous tissues (N) from 12 patients with HCC. **f** The density of bands in **e** was quantified using Image J and relative expression to GAPDH was calculated ($n = 12$). **g** The expression of MTERF3 in the 12 cases of HCC patients was calculated ($n = 12$). **h** TCGA data analysis showed that *MTERF3* expression is positively correlated with poorer overall survival of HCC patients ($n = 363$). Log-rank test for overall survival analysis, student's *t* test for others; *$P < 0.05$ and ***$P < 0.001$.

**MTERF3 participates in regulating cell cycle progression in HCC cells.** Uncontrolled cell cycle is a typical feature of tumor cells[17]. Therefore, we further examined the effect of MTERF3 on the cell cycle progression of HCC cells. HCC-97H and LM3 cells were transfected with siNC or siMTERF3s for 3 days, and cells were collected for flow cytometry analysis. The results showed that MTERF3 knockdown significantly decreased the percentage of cells in G1 phase but increased cells in S-G2/M phase in both HCC-97H and LM3 cells (Fig. 3a, b and

Supplementary Fig. 4a, b). Consistently, western blot results showed that MTERF3 knockdown suppressed the expression of several cell cycle regulators, including CDK2, CDK4, cyclin A, cyclin B1, cyclin E and cyclin D1 (Fig. 3c, d). These results indicate that MTERF3 knockdown inhibits cell cycle progression and causes S-G2/M cell cycle arrest in HCC cells. In contrast, Huh7 cells with stably overexpressed MTERF3 displayed a significantly increase of S phase when compared with controls (Supplementary Fig. 4c, d). Collectively, we

**Table 1 Relationship between MTERF3 expression and clinicopathological features of HCC patients.**

| Variables | MTERF3 expression | | $\chi^2$ | P value |
|---|---|---|---|---|
| | Low | High | | |
| Gender | | | 1.567 | 0.211 |
| Female | 0 | 3 | | |
| Male | 12 | 22 | | |
| Age (years) | | | 0.059 | 0.809 |
| <50 | 2 | 5 | | |
| ≥50 | 10 | 20 | | |
| HBsAg | | | 0.025 | 0.874 |
| Positive | 8 | 16 | | |
| Negative | 4 | 9 | | |
| AFP (ng/mL) | | | 0.131 | 0.717 |
| <20 | 7 | 13 | | |
| ≥20 | 5 | 12 | | |
| Liver cirrhosis | | | 0.059 | 0.809 |
| Yes | 10 | 20 | | |
| No | 2 | 5 | | |
| Tumor number | | | 0.059 | 0.809 |
| Single | 10 | 20 | | |
| Multiple | 2 | 5 | | |
| Tumor size | | | 3.384 | 0.066 |
| <5 cm | 10 | 13 | | |
| ≥5 cm | 2 | 12 | | |
| Tumor differentiation | | | 0.987 | 0.61 |
| Well | 4 | 5 | | |
| Moderate | 7 | 16 | | |
| Poor | 1 | 4 | | |
| TNM stage | | | 3.970 | 0.046 |
| I | 8 | 8 | | |
| II-IV | 4 | 17 | | |

demonstrated that MTERF3 expression is positively correlates with cell cycle progression.

**Suppression of MTERF3 promotes apoptosis in HCC cells.** Besides the cell cycle arrest, an accumulation of sub-G1 cells was also observed from the results of flow cytometry analysis (Fig. 3a, b and Supplementary Fig. 4a, b), which indicated that MTERF3 knockdown may induce apoptosis in HCC cells. To test this, MTERF3 knockdown cells and control cells were stained with Annexin V and analyzed by flow cytometry. The results further verified that MTERF3 knockdown induced a significant increase of Annexin V-positive cells (Fig. 4a, b and Supplementary Fig. 4e, f). Western blot results also showed that MTERF3 knockdown strikingly increased the level of cleaved caspase 3 and PARP1 (Fig. 4c and Supplementary Fig. 4g). We further explored the effects of MTERF3 on non-malignant liver cells, and found that MTERF3 knockdown could not induce a significantly increased of apoptosis, although accompanied with mildly cell proliferation inhibition in LO2 cells (Supplementary Fig. 5).

Given the critical function of MTERF3 in regulating mtDNA transcription, we determined the role of mitochondria-dependent pathway in MTERF3 mediated apoptosis. We found that MTERF3 knockdown induced an increase of cleaved caspase 9, as well as a decrease of mitochondrial membrane potential (Fig. 4c–e). In addition, MTERF3 knockdown induced an increased transcription of pro-apoptotic BH3-only protein *BID* and decreased transcription of anti-apoptotic *Bcl-2* and *Bcl-XL* (Fig. 4f). Moreover, treatment with Z-LEHD-FMK, a specific caspase 9 inhibitor, greatly impaired the apoptosis induced by MTERF3 knockdown (Fig. 4g). Taken together, these results indicate that MTERF3 knockdown-mediated cell apoptosis majorly depends on mitochondria-dependent apoptotic pathway in HCC cells.

**MTERF3 knockdown induces proliferation inhibition and apoptosis in HCC cells via p38 MAPK signaling pathway.** To explore the mechanism by which MTERF3 regulates cell proliferation and apoptosis in HCC cells, we examined several signaling pathways closed association with cell proliferation and apoptosis. Among them, we found that the phosphorylation of p38 but not JNK and ERK was significantly elevated upon MTERF3 knockdown in both HCC-97H and LM3 cells (Fig. 5a and Supplementary Fig. 6). It was known that p38 MAPK signaling pathway plays a critical role in regulating the cell response to extra- and intracellular stress[18]. To address whether p38 MAPK signaling pathway involved in MTERF3-mediated cell proliferation inhibition and apoptosis, a specific siRNA against to p38 or a p38 inhibitor, SB203580, were used to reverse the p38 MAPK activation induced by MTERF3 knockdown in HCC cells (Fig. 5b, c). Cell count assay revealed that suppression of p38 MAPK could significantly restore the proliferation capability of in MTERF3 knockdown cells (Fig. 5d, e). Consistently, p38 MAPK suppression also significantly reversed the apoptosis and PARP1 cleavage induced by MTERF3 knockdown (Fig. 5b, c, f–g and Supplementary Fig. 7). Altogether, these results suggest that p38 MAPK signaling pathway, at least to a large extent, mediates the proliferation inhibition and cell apoptosis upon MTERF3 knockdown-induced stress in HCC cells.

**MTERF3–mediated proliferation inhibition and p38 MAPK activation depends on mitochondrial dysfunction-induced ROS accumulation.** MTERF3 plays multiple important functions in controlling mitochondria steady, such as mtDNA transcription and mitochondrial ribosome biogenesis in mouse and Drosophila cells[13,19]. Therefore, we speculated that MTERF3-mediated mitochondrial abnormality may contribute to p38 MAPK activation. To test the hypothesis, we first detected the effect of MTERF3 on mtDNA transcription in HCC cells. qRT-PCR results revealed that the transcription of most tested mitochondrial genes was increased when MTERF3 knockdown, but mildly inhibited when MTERF3 overexpression (Fig. 6a and Supplementary Fig. 8a, b). However, the mtDNA content and ATP production were significantly reduced in MTERF3 knockdown cells (Fig. 6b, c and Supplementary Fig. 8c, d), which suggests that MTERF3 knockdown resulted in a mitochondrial functional deficiency in HCC cells. To further verify the hypothesis, we evaluated the effects of MTERF3 knockdown on mitochondrial function via measuring oxygen consumption rate (OCR), an important indicator of Oxidative phosphorylation, by using Oxytherm Clark-type electrode. The results revealed that MTERF3 knockdown induced a significant reduction of OCR in basal respiration and maximal respiration but not ATP turnover in HCC-97H cells (Fig. 6d). Mitochondrial dysfunction is usually accompanied with ROS accumulation[20]. Therefore, we determined the total ROS level and mitochondrial ROS accumulation in HCC cells after MTERF3 knockdown. As shown in Fig. 6e, f and Supplementary Fig. 8e–h, compared with controls, MTERF3 knockdown markedly elevated both total cellular ROS and mitochondrial ROS production.

Next, we investigated the role of mitochondrial dysfunction-induced ROS accumulation in MTERF3-mediated cell proliferation inhibition and p38 MAPK activation. HCC-97H or LM3 cells were pre-treated with Trolox, a ROS scavenger, and then were transfected with siMTERF3. Flow cytometry analysis showed that ROS accumulation-induced by MTERF3 knockdown was significantly eliminated by Trolox treatment (Fig. 6g, h). Meanwhile,

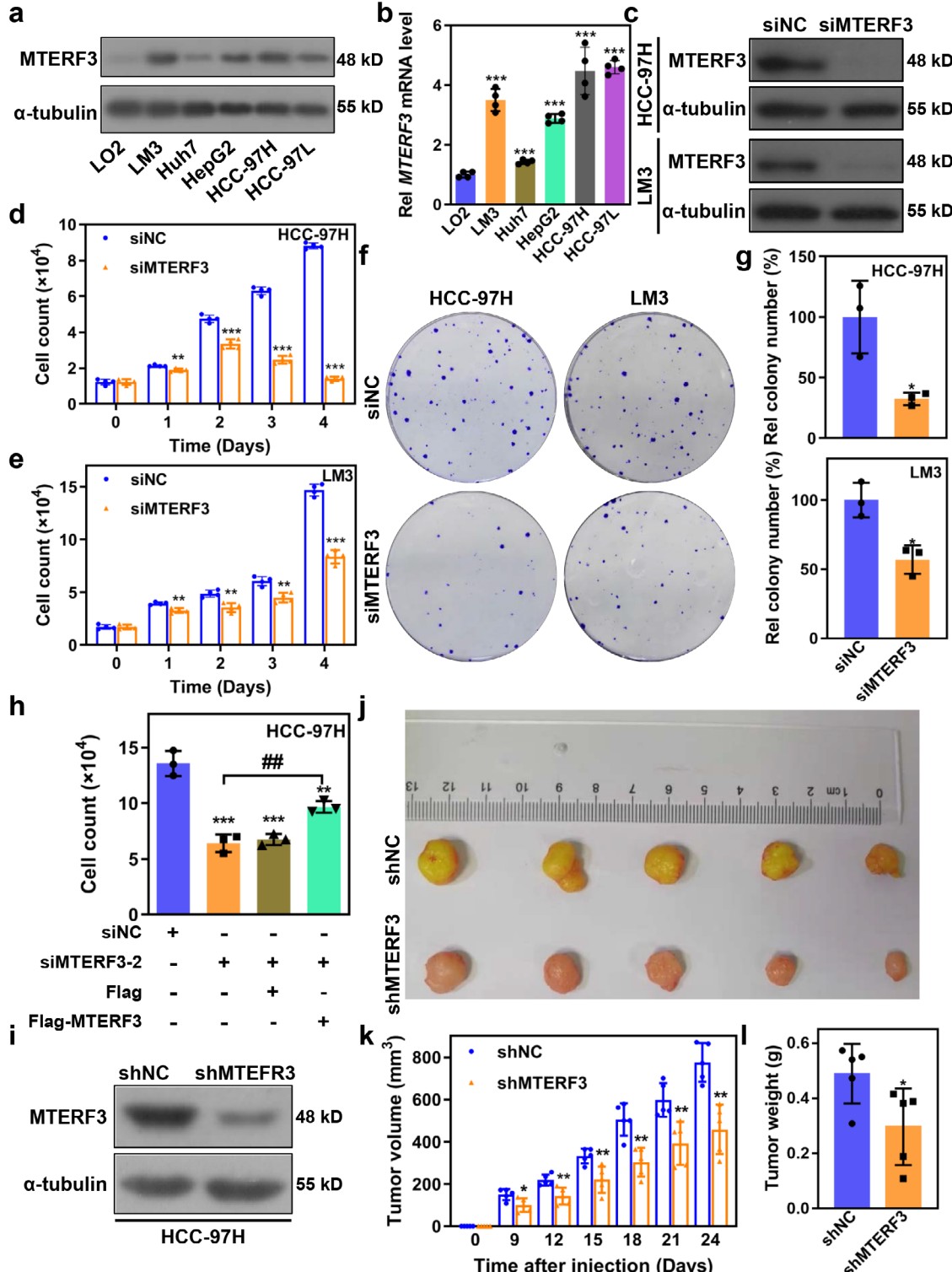

**Fig. 2 MTERF3 knockdown inhibits cell proliferation of HCC cells in vitro and in vivo.** Western blot (**a**) or qRT-PCR (**b**) was used to examine the expression of MTERF3 in normal hepatocytes LO2 cells and HCC cells ($n = 4$). **c** HCC-97H or LM3 cells were transfected with indicated siRNA for 3 days, and the cell lysates were used for detecting the expression of MTERF3 by western blot. Cell count was used to analyze the proliferation of HCC-97H (**d**) or LM3 (**e**) cells after MTERF3 knockdown at indicated times ($n = 3$). **f** Colony formation assay to detect the cell proliferation ability of HCC cells after MTERF3 knockdown. **g** The colony numbers were counted using Image J software and relative colony number were calculated ($n = 3$). **h** HCC-97 cells transfected with a siRNA targeting 3'-translated region of *MTERF3* mRNA together with indicated vectors were transfected into HCC-97H cells, and cell count was used to analyze the expression of MTERF3 and its effects on cell proliferation ($n = 3$). **i** Western blot analysis for MTERF3 expression in HCC-97H cells with stably MTERF3 knockdown and controls. **j** Image of subcutaneous xenograft tumor formed by HCC-97H cells with stably MTERF3 knockdown or controls in nude mice ($n = 5$). Tumor growth (**k**) and statistical analysis of the weight of tumors (**l**) in **j** ($n = 5$). Data are shown as mean ± standard deviations. Student's *t* test; *$P < 0.05$, **$P < 0.01$ and ***$P < 0.001$ compare to siNC; ##$P < 0.01$.

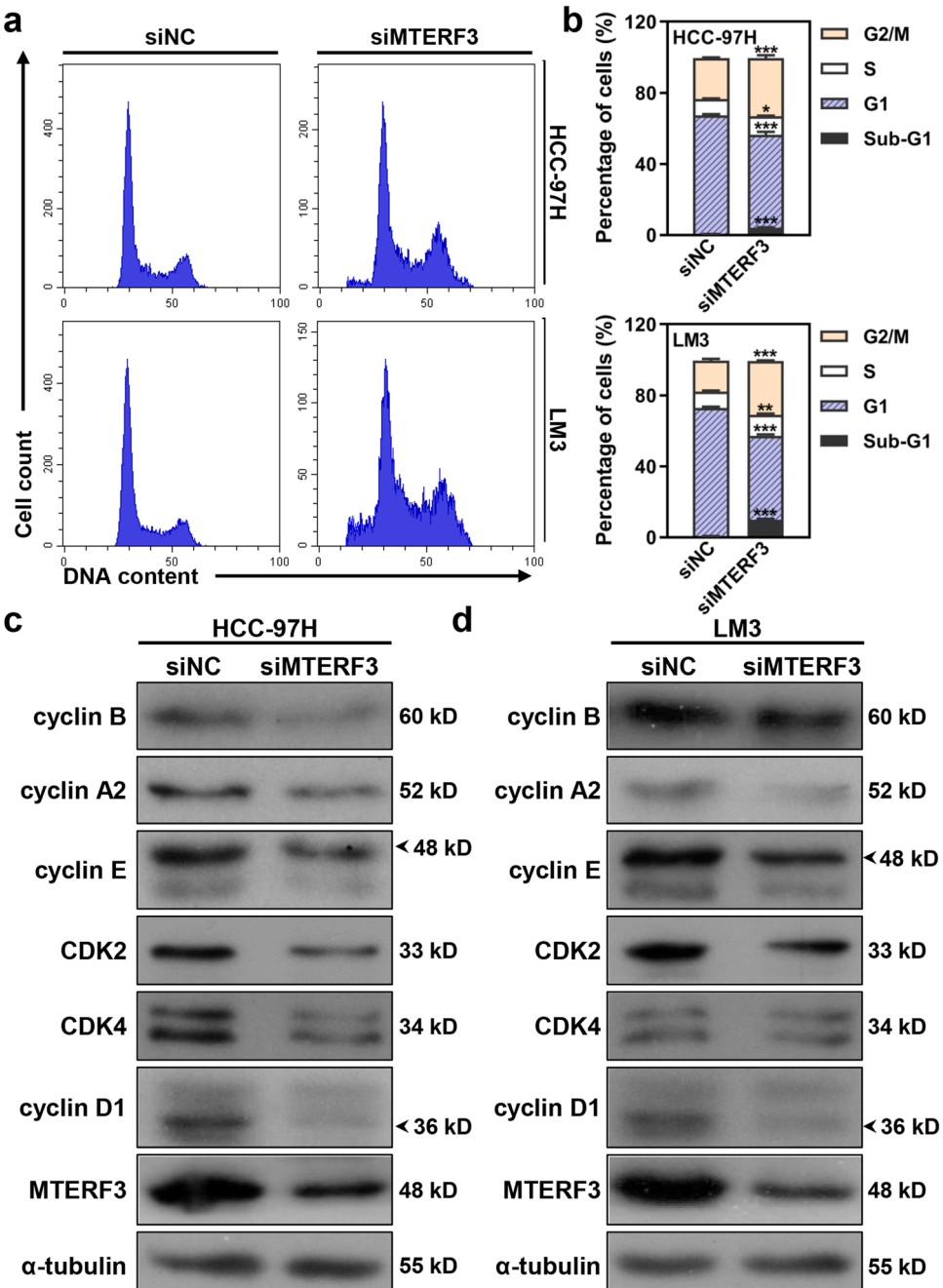

**Fig. 3 MTERF3 knockdown induces cell cycle arrest at S-G2/M phase. a** Cell cycle profile of HCC cells after transfected with siNC or siMTERF3. **b** Cell cycle distribution in **a** was calculated (*n* = 3). **c** HCC cells were transfected with indicated siRNA for 3 days, and cell lysates were used to analyze the expression of cell cycle related proteins. All experiments were performed at least three times. Data are shown as mean ± standard deviations. Student's *t* test; *P < 0.05, **P < 0.01 and ***P < 0.001.

MTERF3 knockdown -induced cell proliferation inhibition and p38 activation were also markedly rescued when cells were treated with Trolox (Fig. 6i, j). Taken together, these results indicate that MTERF3-mediated cell proliferation inhibition and p38 activation is dependent on mitochondrial dysfunction-induced ROS accumulation.

## Discussion
In this study, we reveal that MTERF3 is strikingly enhanced in HCC tissues and its higher expression positively correlates with the TNM stage and poorer prognosis of HCC patients. MTERF3 knockdown causes an inhibition of mtDNA transcription and an

impaired Oxidative phosphorylation, which elicits mitochondrial dysfunction and ROS accumulation, leading to p38 MAPK activation, thereby suppressing the proliferation of HCC cells via inducing S-G2/M cell cycle arrest and apoptosis (Fig. 7).

MTERF3 contains a classical mitochondrial transcription termination factor motif, thus it is originally defined as a terminate factor of mitochondrial transcription. Indeed, the original finding found that MTERF3 is a negative regulator of mitochondrial transcription, and MTERF3 can bind to the promoter region of mtDNA and inhibit the transcription initiation in mouse cells[13]. Besides the negative role in regulating mitochondrial transcription, MTERF3 also was found to regulate mtDNA replication,

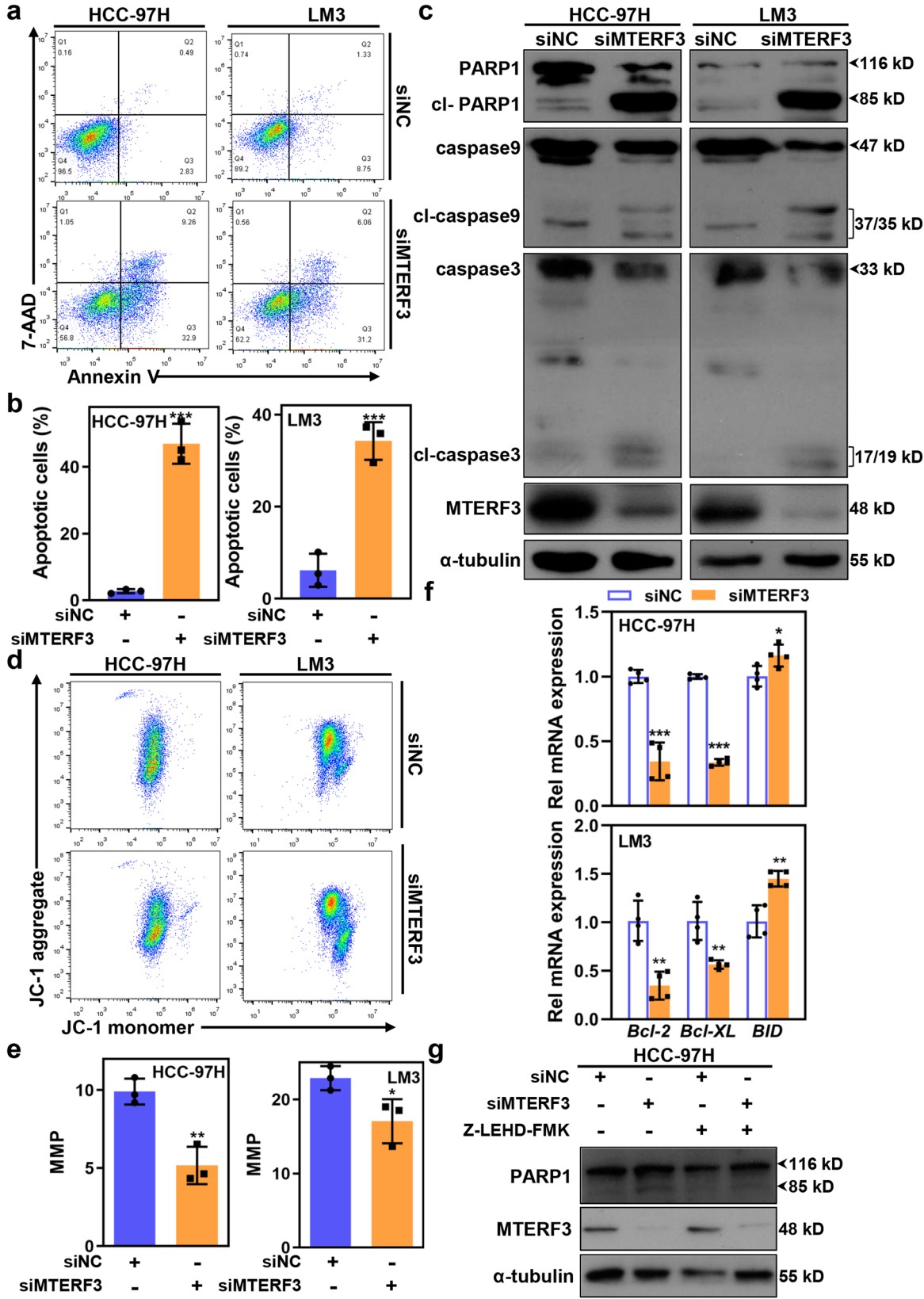

mitochondrial ribosome biogenesis and protein translation in Drosophila[19,21]. These results indicate that, although MTERF3 is the most conserved mitochondrial transcription termination factor, it seemingly plays different roles in regulating mitochondrial function in different organisms. Nevertheless, MTERF3 depletion induces mitochondrial dysfunction and deficiency of

development, resulting severe respiratory deficiency and embryonic lethality[13,19]. We found that MTERF3 knockdown caused a significantly increase of mitochondrial transcription in HCC cells, which suggests that MTERF3 is an evolutionarily conserved negative regulator of mitochondrial transcription and critical for sustaining mitochondrial dynamic-steady.

**Fig. 4 MTERF3 knockdown induces apoptosis in HCC cells via mitochondria dependent apoptotic pathway. a** Annexin V/7-AAD staining to analyze cell apoptosis of HCC cells after transfected with siNC or siMTERF3. **b** Cell apoptosis in **a** was calculated ($n = 3$). **c** HCC cells were transfected with indicated siRNA for 3 days, and cell lysates were used to analyze the expression of apoptosis related proteins. **d** JC-1 staining to examine the mitochondrial membrane potential (MMP) of HCC cells after transfected with siNC or siMTERF3. **e** The MMP of HCC cells in **d** was calculated ($n = 3$). **f** qRT-PCR to determine the expression of apoptosis related Bcl-2 family genes ($n = 4$). **g** siNC or siMTERF3 transfected HCC-97H cells were treated with 20 μM Z-LEHD-FMK, and cell lysates were used to analyze the expression of indicated proteins. All experiments were performed at least three times. Data are shown as mean ± standard deviations. Student's $t$ test; *$P < 0.05$, **$P < 0.01$ and ***$P < 0.001$.

However, despite an elevated mtDNA transcription was observed upon MTERF3 knockdown, it seems a little paradoxical that depletion of MTERF3 induced a reduction of mtDNA abundance and ATP production, as well as a significant decrease in basal OCR and maximal OCR. As semiautonomous organelles, the dynamics and function of mitochondria are strictly controlled by nuclear genome. We speculated that MTERF3 knockdown -induced elevation of mtDNA transcription may feedback to nuclear genome, and causes a disorder in mitochondrial dynamics and mitochondrial dysfunction. Consistently, previous study also found that overexpression of mitochondrial transcription factor A could stimulate the transcription of mtDNA but led to a mildly decrease in abundance of full-length mtDNA[22]. Nevertheless, we also can not eliminate the possibility that MTERF3 has another functions independent on its role in mitochondrial transcription termination.

p38 MAPK pathway plays a dual role in regulating cell survival, and it can either promote cell survival or facilitate cell death depending on the type of cells and stimuli[23]. For example, p38 activity is usually upregulated in several solid tumors and plays an anti-apoptotic and proliferative effects[23–28]. circFAM120B as a tumor suppressor could inhibit cell proliferation and metastasis via suppressing p38 signaling in esophageal squamous cell carcinoma[29]. However, numerous studies have found that, upon extra- or intracellular stress including ROS accumulation, ionizing radiation or chemotherapeutic drugs treatment, etc., p38 MAPK activity is elevated and causes cell arrest or cell death[30–35]. In our study, we also showed that MTERF3 knockdown induces mitochondrial dysfunction and ROS accumulation, which promotes p38 MAPK-dependent cell proliferation inhibition, cell cycle arrest and apoptosis.

In conclusion, we found that MTERF3 is frequently upregulated in HCC tumor tissues, and its higher expression positively correlated with TNM stage and poor prognosis of HCC patients. Targeting MTERF3 induces mitochondrial transcription inhibition, mitochondrial dysfunction and ROS accumulation, activating p38 MAPK pathway and inhibiting HCC cells proliferation, which suggests that MTERF3 may serve as a promising biomarker used for HCC prognosis and targeted treatment.

## Methods

**Patients and specimens.** A total of 50 patients with primary HCC from The First Affiliated Hospital of Wenzhou Medical University (Zhejiang, China) were enrolled, and their tumor tissues and corresponding non-cancerous tissues were collected and used for analyzing the expression of MTERF3 by qRT-PCR or western blot. All patients signed informed consent before surgical resection. The study was approved by the Research Ethics Committee of Wenzhou Medical University (0000-0002-3120-3121).

**Cell culture, transfection and drug treatment.** Human HCC cell lines, including HCC-97H, HCC-97L, LM3, Huh7 and HepG2, and a normal hepatocytes cells LO2 were provided by L.X. (Wenzhou Medical University) and all cell lines were characterized by DNA fingerprinting and isozyme detection. HCC cells were cultured in DMEM medium (Gibco, California, USA) supplemented with 10% (v/v) fetal bovine serum (FBS, Gibco). LO2 cells were cultured in RPMI 1640 medium (Gibco) supplemented with 10% FBS. All cells were cultured in a humidified incubator under 5% $CO_2$ at 37 °C.

Nonsense control siRNA (siNC: 5'-UUCUCCGAACGUGU-CACGUUU-3') and siRNAs specific targeting MTERF3 (siM-TERF3: 5'-GCCCAACAGAUACCCAGAU-3'; siMTERF3-1: 5'-GGGUAUAGAGGAUAACCAAUU-3'; siMTERF3-2: 5'-AAUGCCUCGGGUCUCAACUUU-3') were synthesized by GenePharma (Shanghai, China), and transfection was conducted with Lipofectamine 3000 (Life Technologies Inc., Massachusetts, USA) according to the manufacturer's protocols.

For p38 inhibitor (SB203580, MedChemExpress, New Jersey, USA) or ROS scavenger (Trolox, TOPSCIENCE, Shanghai, China) treatment, HCC cells were pretreated with 20 μM SB203580 or 900 μM Trolox for 1 h. After siRNA transfection, cells were incubated with medium containing SB203580 or Trolox for 72 h. For caspase 9 inhibitor (Z-LEHD-FMK, BD Biosciences, New Jersey, USA) treatment, siRNA transfected HCC-97H cells were treated with 20 μM Z-LEHD-FMK for 72 h.

**RNA extraction and quantitative real-time PCR (qRT-PCR).** Total RNA of the tissues and cells were extracted using TRIzol reagent (Life Technologies Inc.) according to the manufacturer's introductions. Reverse Transcriptase Reaction Kit (Vazyme, Nanjing, China) were used to synthesize cDNA. qRT-PCR was performed on a Bio-Rad CFX 96 Touch with SYBR GREEN Master Mix (Vazyme). The primer sequences for qRT-PCR were shown as follows: 5'- GCGCCCTGCAAATTGTAA -3' (forward) and 5'- CCATTGCGCTTCATTTGGA -3' (reverse) for *MTERF3*, 5'- CCACCTCTAGCCTAGCCGTTTA-3' (forward) and 5'-GGGTCATGATGGCAGGAGTAAT-3' (reverse) for *ND1*, 5'-CAAACAATGTTCAACCAGTAACCACTAC-3' (forward) and 5'-ATATACTACAGCGATGGCTATTGAGGA-3' (reverse) for *ND6*, 5'-TAGCCATACACAACACTAAAGGACGA-3' (forward) and 5'-GGGCATTTTTAATCTTAGAGCGAAA-3' (reverse) for *ATP6*, 5'-TAGAGGAGCCTGTTCTGTAATCGAT-3' (forward) and 5'-CGACCCTTAAGTTTCATAAGGGCTA-3' (reverse) for *RNR1*, 5'-ATCACTCGAGACGTAAATTATGGCT-3' (forward) and 5'-TGAACTAGGTCTGTCCCAATGTATG-3' (reverse) for *Cyt B*, 5'-AAGCGGTCCCGTGGATAGA-3' (forward) and 5'-TCCGGTATTCGCAGAAGTCC-3' (reverse) for *Bcl-2*, 5'-GGAGCTGGTGGTTGACTTTCT-3' (forward) and 5'-CCGGAAGAGTTCATTCACTAC-3' (reverse) for *Bcl-XL*, 5'-ACCCTAGAGACATGGAGAAG-3' (forward) and 5'-AGC-TATCTTCCAGCCTGTCT-3' (reverse) for *BID*, and 5'-TGCGTTACACCCTTTCTTGACA-3' (forward) and 5'-GCAAGGGACTTCCTGTAACAATG-3' (reverse) for *β-actin*. $2^{-\Delta\Delta Ct}$ was used to calculate relative mRNA levels and *β-actin* was used as internal control.

**Western blot.** Cells were collected and lysed with RIPA buffer and equal amounts of total protein lysates were used for SDS-

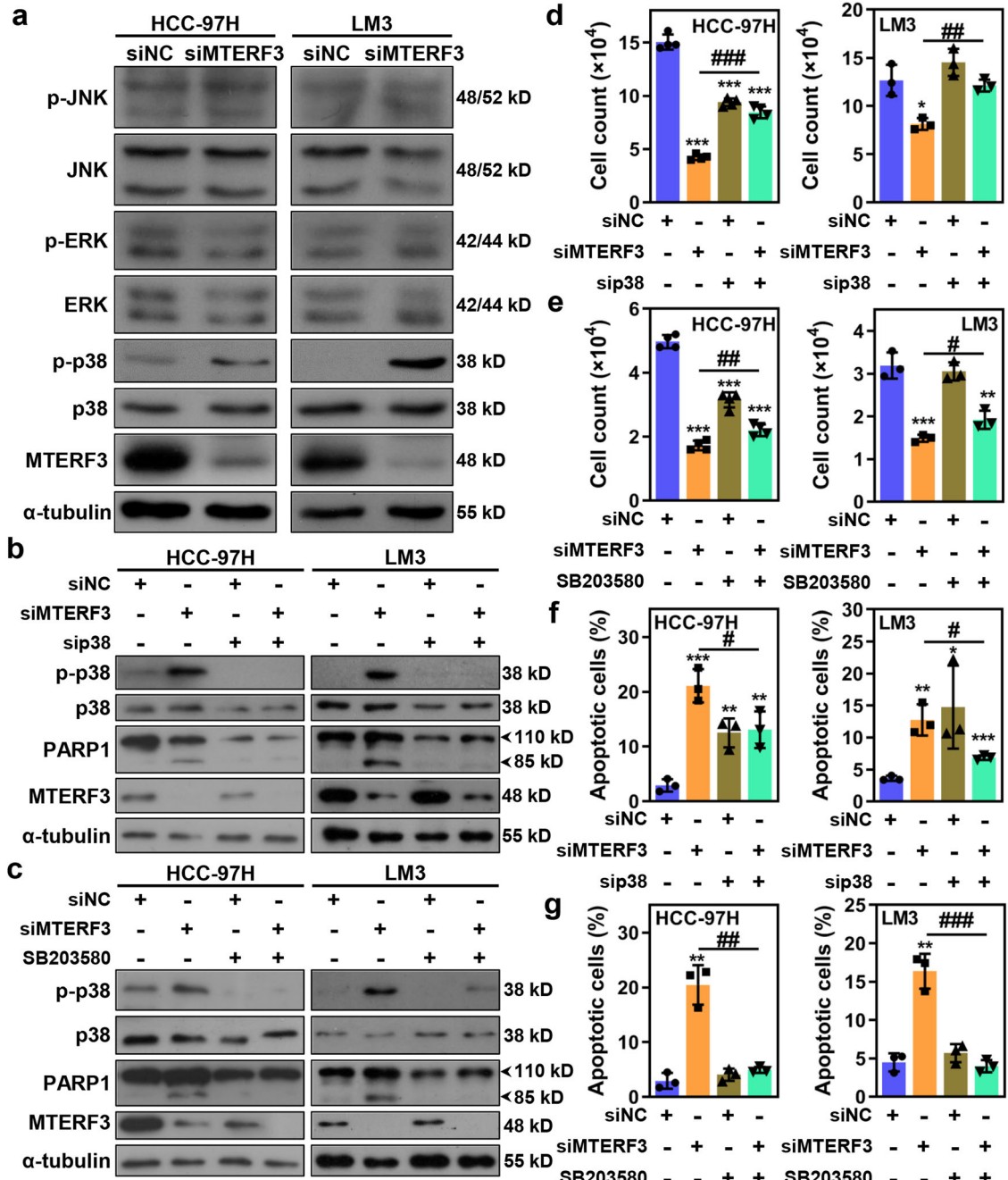

**Fig. 5 p38 MAPK activation is required for MTERF3-mediated cell proliferation inhibition and apoptosis in HCC cells. a** HCC-97H or LM3 cells were transfected with indicated siRNA for 3 days, and cell lysates were used to detect the expression of MAPK pathway proteins. Western blot to assess the effect of p38 siRNA transfection (**b**) or p38 inhibitor SB203580 treatment (**c**) on MTERF3-induced p38 activation and PARP1 cleavage in HCC-97H or LM3 cells. Cell count to analyze the effect of p38 siRNA transfection (**d**) or p38 inhibitor SB203580 treatment (**e**) on MTERF3-induced cell proliferation inhibition in HCC-97H or LM3 cells ($n = 4$ for HCC-97H and $n = 3$ for LM3). Annexin V/7-AAD assay to examine the effect of p38 siRNA transfection (**f**) or p38 inhibitor SB203580 treatment (**g**) on MTERF3-induced apoptosis in HCC-97H or LM3 cells ($n = 3$). Data are shown as mean ± standard deviations. Student's $t$ test; $^*P < 0.05$, $^{**}P < 0.01$ and $^{***}P < 0.001$ compare to siNC; $^\#P < 0.05$, $^{\#\#}P < 0.01$ and $^{\#\#\#}P < 0.001$.

polyacrylamide gel electrophoresis and immunoblotting[36]. Then antibodies information was listed in Supplementary Table 1.

**Cell proliferation, cell cycle and apoptosis analysis.** For the cell count, 15000 cells were seeded into 24-well plates. After transfection with indicated siRNA, cells were collected and counted every 24 h using cell counting chamber. For colony formation assay, 500 cells were plated in 6-well plate and cultured for

10 days. Colonies were counted manually after staining with 0.1% crystal violet. Colony number was analyzed by Image J.

For cell cycle analysis, siRNA transfected HCC cells were collected and fixed with 70% ethanol /30% PBS for overnight at −20 °C. The cells were subsequently washed once with PBS, and incubated in propidium iodide (PI) buffer (60 μg/ml PI and 0.1 mg/ml RNase A) for 45 min at room temperature. Cells were analyzed by CytoFLEX (Beckman Coulter, Pasadena, USA) and at least 10,000 cells per condition were collected. Cell cycle profiles

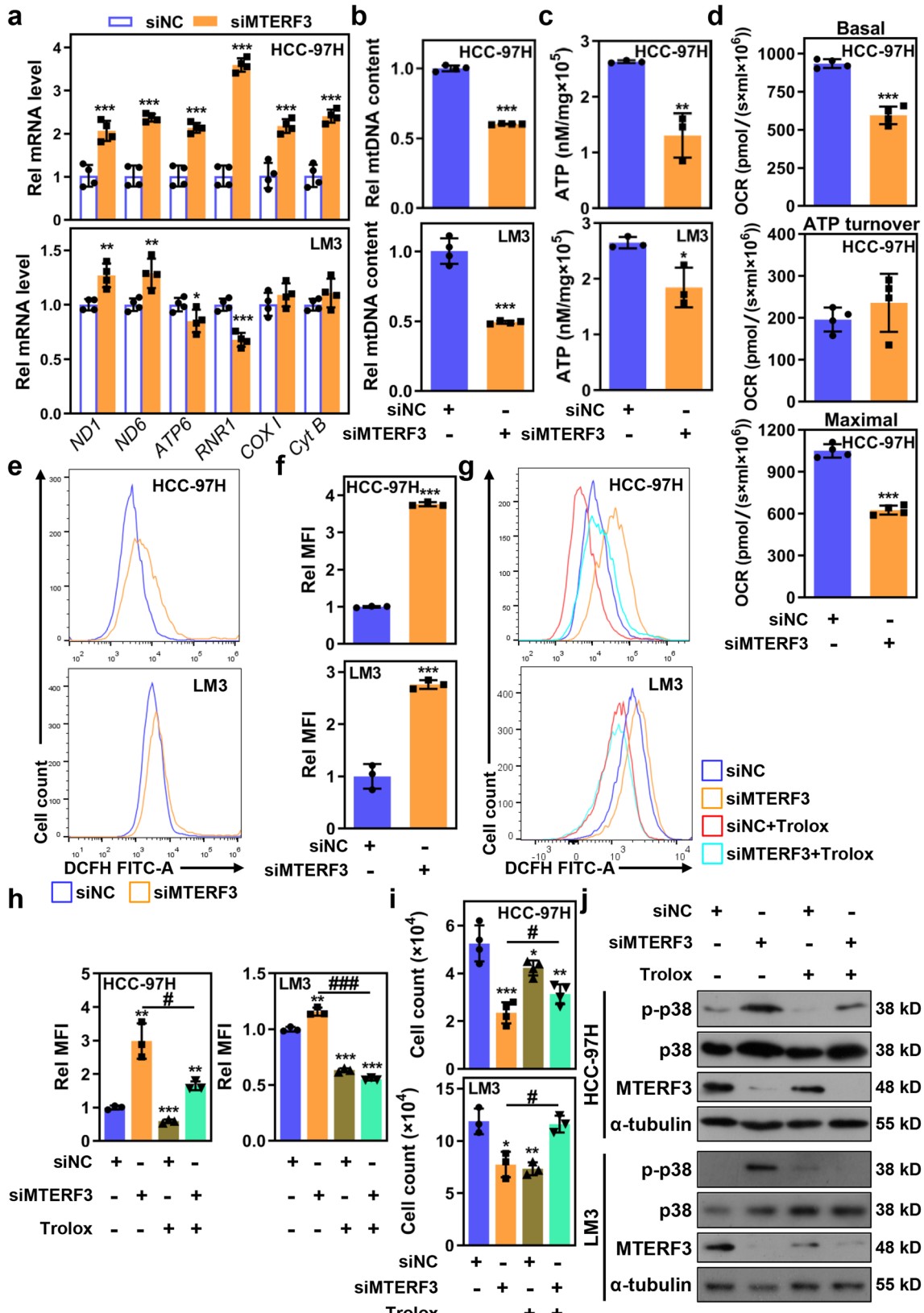

were processed and analyzed using FlowJo_v10.6.2. The gating strategy has been displayed in Supplementary Fig. 9a.

For apoptosis analysis, siRNA transfected HCC cells were collected and were stained with Annexin V-PE and 7-AAD as described in the Annexin V FITC Apoptosis Detection Kit (KeyGen Biotech, Nanjing, China) and data was acquired using

CytoFLEX and analyzed using FlowJo_v10.6.2. The gating strategy has been displayed in Supplementary Fig. 9b.

**Mitochondrial membrane potential detection.** JC-1 Assay Kit (Beyotime, Shanghai, China) was used to detect the

**Fig. 6 MTERF3–mediated p38 MAPK activation and cell proliferation inhibition depends on mitochondrial dysfunction-induced ROS accumulation.**
**a** qRT-PCR to examine mtDNA transcription in HCC-97H or LM3 cells transfected with indicated siRNA ($n = 4$). **b** qRT-PCR to examine the mtDNA content in HCC-97H or LM3 cells transfected with indicated siRNA ($n = 4$). **c** ATP production was detected in HCC-97H or LM3 cells transfected with indicated siRNA ($n = 3$). **d** The effects of MTERF3 knockdown on basal respiration, ATP turnover and maximal respiration were examined using Oxytherm Clark-type electrode ($n = 4$). **e, f** The ROS level in HCC-97H or LM3 cells transfected with indicated siRNA was analyzed, and the relative ROS level in **e** was calculated ($n = 3$). **g, h** siNC or siMTERF3 transfected HCC-97H or LM3 cells were pretreated with ROS scavenger Trolox (0.9 mM), and the effect of which on MTERF3 knockdown-induced ROS accumulation was evaluated ($n = 3$). **i** Cell count to analysis of Trolox treatment on MTERF3 knockdown-induced cell proliferation inhibition ($n = 4$ for HCC-97H and $n = 3$ for LM3). **j** Western blot to assess the effect of Trolox treatment on MTERF3 knockdown-induced p38 activation. All experiments were performed at least three times. Data are shown as mean ± standard deviations. Student's $t$ test; *$P < 0.05$, **$P < 0.01$ and ***$P < 0.001$ compare to siNC; #$P < 0.05$ and ###$P < 0.001$.

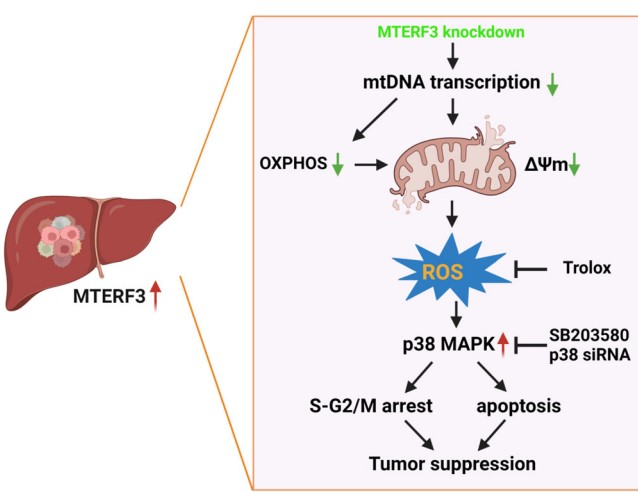

**Fig. 7 The model of MTERF3 regulates HCC progression.** MTERF3 is strikingly enhanced in HCC tissues. MTERF3 knockdown inhibits mtDNA transcription and impairs Oxidative phosphorylation (OXPHOS), which elicits a decrease of mitochondrial dysfunction-mediated mitochondrial membrane potential (△Ψm) and ROS accumulation, leading to p38 MAPK activation, thereby suppressing the proliferation of HCC cells via inducing S-G2/M cell cycle arrest and apoptosis. ROS scavenger Trolox treatment or p38 inhibition could reverse the phenotypes.

mitochondrial membrane potential according to the manufacturer's instructions. In brief, cells were harvested and incubated with JC-1 dye solution for 20 min at 37 °C. After washed twice with JC-1 buffer, the intensities of green and red fluorescence were examined by flow cytometry. The fluorescent intensity was analyzed by Flowjo software. The gating strategy has been displayed in Supplementary Fig. 9c.

**mtDNA copy number assay.** Genomic DNA was extracted from HCC cells using QIAamp DNA mini kit (Qiagen, Hilden, Germany). qRT-PCR was used to amplify the mitochondrial gene *MT-mito* and nuclear gene *beta-2 microglobulin* (*B2M*). Relative mtDNA copy number was calculated by mtDNA/nDNA. The primers were shown as follows: 5'-CACTTTCCACACAGA-CATCA-3' (forward) and 5'- TGGTTAGGCTGGTGTTAGGG-3' (reverse) for *MT- mito*, and 5'- TGTTCCTGCTGGGTAGCTCT-3' (forward) and 5'-CCTCCATGATGCTGCTTACA-3' (reverse) for *B2M*.

**Intracellular ROS assay.** Reactive Oxygen Species Assay Kit (Beyotime) was used to detect intracellular ROS by fluorescence intensity according to the instructions. In brief, cells were harvested after siRNA transfection or Trolox treatment, and were incubated with 10 μM 2′, 7′-dichlorofluorescein diacetate (DCFH-DA) for 30 min at 37 °C. Flow cytometry was used to analyze the

DCF fluorescence intensity. The gating strategy has been displayed in Supplementary Fig. 9d. Relative mean fluorescence intensity (MFI) of DCFH-DA was determined as a ratio of the detected fluorescence signals of experimental treatment groups to controls.

**Mitochondrial ROS assay.** Mitochondrial ROS was accessed by using MitoSOX Red Kit (HY-D1055, MedChemExpress). Cells transfected with indicated siRNA were incubated with 10 μM MitoSOX Red for 30 min at room temperature. Then the cells were rinsed twice using serum-free cell culture solution, and images were captured using a NIKON confocal microscope (Nikon A1, Tokyo, Japan) immediately. The fluorescent intensity was evaluated by Image J software (National Institutes of Health, USA).

**Oxygen consumption rate (OCR) assay.** The OCR was determined with an Oxytherm Clark-type electrode (Hansatech Instruments, Norfolk, UK). Briefly, stably MTERF3 knockdown HCC-97H cells or control cells were harvested from 10 cm dish and washed with TDS (25 mM Tris-base, 137 mM NaCl, 10 mM KCl, 0.7 mM $Na_2HPO_4$, 10% FBS, pH 7.4–7.5) and then centrifugation at 1000 rpm for 5 min. The cells were resuspended in TD (25 mM Tris-base, 137 mM NaCl, 10 mM KCl, 0.7 mM $Na_2HPO_4$, pH 7.4–7.5) and added to the test chamber to record the basal respiratory oxygen consumption. ATP production or maximal oxygen consumption rate was calculated in response to oligomycin (1 μL, 0.1 mg/mL, Sigma-Aldrich, Missouri, USA) or trifluoromethoxy carbonylcyanide phenylhydrazone (FCCP, 1 μL, 0.1 mM, Sigma-Aldrich), respectively. All values were normalized by cell numbers.

**Subcutaneous xenograft experiments.** All animal experiments were approved by the Ethics Committee for Laboratory Animals of the Wenzhou Medical University. In this study, four weeks old BALB/c female nude mice (3–4 weeks old) were randomly divided into two groups ($n = 5$ per group). A total of $5 \times 10^6$ MTERF3 stably knockdown HCC-97H cells or control cells were injected subcutaneously into the right axillary fossa of nude mice. Tumors were measured per three days by a caliper, and tumor volume (V) was calculated using the equation V= (length × width$^2$) /2.

**Statistics and reproducibility.** Statistical analyses were performed using Graphpad Prism 7 and was displayed as mean ± standard deviations with at least three independent experiments. Kaplan–Meier method and logrank test were used to estimate the probability of differences in overall survival, relapse-free survival and progression-free survival. The Pearson's $\chi^2$ test was used to evaluate the relationship between MTERF3 expression and clinicopathological parameters. Students' two-tailed $t$ test was used to assess the significance of different groups and $P < 0.05$ was considered statistically significant.

**Reporting summary**. Further information on research design is available in the Nature Portfolio Reporting Summary linked to this article.

## Data availability

All data supporting the study are available within the paper and/or the Supplementary Information or available from the corresponding authors on reasonable request. Uncropped and unedited blot images are available in Supplementary Fig. 10. All numerical source data for graphs and charts are presented in Supplementary Data.

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

## Acknowledgements

We thank the Scientific Research Center of Wenzhou Medical University for consultation and instrument availability that supported this work. This work was supported by Medical and Health Science and Technology Program of Zhejiang Province to Z.Z. (Grant No. 2023RC202), the Basic Research Project of Wenzhou Municipal Science and Technology Bureau to T.W. (Grant No. Y2023894), Zhejiang Provincial Natural Science Foundation of China to Z.D. (Grant No. LY21C070004), National Natural Science Foundation of China to L.H. (Grant No. 32000622), Key Discipline of Zhejiang Province in Medical Technology (First Class, Category A) (Grant No. 437601607). The model figure of this manuscript was generated using BioRender (www.biorender.com).

## Author contributions

All authors contributed to the study conception and design. Material preparation, data collection and analysis were performed by Z.Z., Y.Z., H.Y., L.H. and Z.D. Resources were prepared by T.W., J.L., L.X., C.D. and L.W. The first draft of the manuscript was written by Z.D. and all authors commented on previous versions of the manuscript. All authors read and approved the final manuscript.

## Competing interests

The authors declare no competing interests.

## Ethics approval and consent to participate

All aspects of this study were approved by Institutional Research Ethics Committee of Wenzhou Medical University.
