## [Peer Review File · Communications Biology]

Reviewers' comments:

Reviewer #1 (Remarks to the Author):

In their manuscript "Suppressing MTERF3 inhibits proliferation of human hepatocellular carcinoma via ROS mediated p38 MAPK activation" Zhihai Zheng and colleagues investigate the correlation between MTERR3 expression and the outcome of HCC patients. They show that MTERR3 displays a negative correlation with prognosis of HCC patients and MTERR3 knockdown inhibits cell proliferation of HCC cells in vitro and in vivo. They describe a mechanism in which MTERF3-mediated cell proliferative inhibition is dependent on mitochondrial dysfunction-induced ROS accumulation. The concept is interesting. However, the data are currently not supporting the author's claims. Overall, there are several major concerns that prevents the manuscript in its current form from being published in Communications Biology.

Major concerns:

- (1) The authors should also perform MTERF3-overexpression assay to provide more convincing results in figure 2-6, not only knockdown assay using siMTERF3.
- (2) For all data, statistical test used should be indicated in the legends.
- (3) Fig. 2A-G: All experiments should be performed at least three times.
- (4) MTERF3 expression should be analyzed by western blot to demonstrate effect of protein knockdown in Figure 2H-J.
- (5) Fig. 5C: The phosphorylation of p38 was not significantly elevated upon MTERF3 KD in HCC-97H cells with p38 siRNA transfection, why?
- (6) Fig. 5C, 5D and 6J: Why not carry out the experiments using LM3 cells.
- (7) Showing individual values is required to display variation in experiments.

Minor comments:

- a) Fig. 2J: Labeling should be improved and match the Figure legend.
- b) The title of Fig. 5 should be improved.

Reviewer #2 (Remarks to the Author):

Hepatocellular carcinoma (HCC) is the most common type of primary liver cancer, which accounts for about 75% to 85% of all liver cancer cases. In this manuscript, Zheng et al. investigate the role of mitochondrial transcription termination factor 3 (MTERF3) in HCC progression. By analyzing MTERF3 expression level in HCC tumor tissues, they found a high expression of MTERF3 correlates with tumor progression in HCC patients. Knockdown of MTERF3 leads to mitochondria dysfunction, ROS accumulation and activation of p38 MAPK signaling pathway, which finally induces cell cycling arrest, apoptosis and HCC cell proliferation inhibition. Overall, these results are interesting and identification of MTERF3 in HCC progression could provide a potential target for HCC cancer treatment. However, there are several points that need to be addressed:

Major

1. Most of the experiments in this manuscript depends on siRNA or shRNA knockdown. As siRNA has strong off-target effect, usually at least two siRNA/shRNA sequence and rescue experiment are needed to exclude off-target effect, especially for some key experiments. This is the standard for knockdown experiments. The authors provide two siRNA sequence data for proliferation and p38 signaling experiments, that is good. However, they should provide data for other key experiments, like Figure 2H-I, Figure 3A-B and Figure 6C-F.
2. One big gap of this manuscript is that how MTERF3 regulates ROS. The authors claim that MTERF3-mediated p38 MAPK activation depends on mitochondrial dysfunction-induced ROS accumulation and knockdown of MTERF3 decreases mitochondria DNA level and ATP generation. In Figure 6A-B, the authors show that knockdown of MTRF3 increases mitochondria gene expression, this is not consistent with Figure 6C which shows a decrease of mtDNA contents. Can the authors explain this? As mitochondria is one major organelle for ROS production, ROS leakage happens from ETC complex or mitochondrial respiration. So the authors should check ETC protein levels by western blot or

mitochondrial respiration by seahorse experiment. This will help explain the ROS accumulation induced by MTRF3 knockdown and makes this part much stronger.

3. Following point 2, the authors detect ROS level here by DCF staining (2',7'-dichlorofluorescein diacetate (DCFH-DA)), which is a commonly used ROS dye. However, a lot of concerns have been raised about this ROS detection methods. Besides, the DCF intensity not only reflects oxidation of this dye but also uptake of this dye into the cell. As ROS regulation by MTRF3 is an important part in this manuscript for the regulation mechanism, the authors should provide more solid experimental evidence for ROS detection, like using other ROS detection kit or genetically encoded ROS reporter. It would be good to detect mitochondria ROS by mitoSOX, as the model here is mitochondrial dysfunction leads to ROS accumulation for P38 signaling regulation.

Minor

1. In Materials and Methods, the authors should give details of bioinformatics analysis for Figure 1A, B and H.

2. Does MTRF3 knockdown also affect the proliferation of normal cells, as it is a key regulator for mitochondrial gene transcription.

3. Some words in the text need to be changed to be more accurate. For example, in line 50, 'Silencing of MTRF3 51 induces mitochondrial dysfunction' should be 'Knockdown of MTRF3 51 induces mitochondrial dysfunction.'

Thank you very much for the review of our manuscript entitled “Suppressing MTERF3 inhibits proliferation of human hepatocellular carcinoma via ROS mediated p38 MAPK activation” by Zheng et al (COMMSBIO-23-1380A). We have carefully read the valuable comments and suggestions. We are grateful to the editors and reviewers for the constructive suggestions that have improved the quality of the manuscript. The point-to-point responses are presented below.

Referee #1 (Remarks to the Author):

Major concerns:

1. The authors should also perform MTERF3-overexpression assay to provide more convincing results in figure 2-6, not only knockdown assay using siMTERF3.

Response: As the reviewer suggested, we constructed a stably MTERF3 overexpressed (OE) HuH7 cells, and examined the effects of MTERF3 OE on cell proliferation and cell cycle progression. The results showed that MTERF3 OE significantly promoted the proliferation and cell cycle progression in HuH7 cells (Figure S3 and S4C-D). Meanwhile, we also found that MTERF3 OE strikingly inhibited mtDNA transcription (Figure S8B). We displayed these results in revised supplemental data.

(2) For all data, statistical test used should be indicated in the legends.

Response: As the reviewer suggested, we had added the statistical information in the revised figure legends.

(3) Fig. 2A-G: All experiments should be performed at least three times.

Response: In fact, all experiments were performed at least three times in Figure 2A-G, and the displayed data were shown as mean \pm standard deviations. We are regretted that

we missed error bars in Figure 2E. We remade the figure and displayed it in the revised manuscript.

(4) MTERF3 expression should be analyzed by western blot to demonstrate effect of protein knockdown in Figure 2H-J.

Response: We constructed a stably MTERF3 knockdown in HCC-97H cells, and verified that MTERF3 can be effectively suppressed by MTERF3 shRNA, and the results were displayed in Figure S2. As the reviewer suggested, we displayed another western blot result in Figure 2H in revised manuscript.

(5) Fig. 5C: The phosphorylation of p38 was not significantly elevated upon MTERF3 KD in HCC-97H cells with p38 siRNA transfection, why?

Response: The reviewer made a good point. We performed the experiment repeatedly. We found that both p38 and p38 phosphorylation was significantly decreased after p38 siRNA transfection (Figure 5B). We added the results in revised manuscript.

(6) Fig. 5C, 5D and 6J: Why not carry out the experiments using LM3 cells.

Response: The reviewer made a good suggestion. We performed the experiments in LM3 cells. We found that, similar to the results in HCC-97H, p38 siRNA transfection and p38 inhibitor treatment could significantly reverse the p38 signaling activation induced by MTERF3 knockdown in LM3 cells (Figure 5B-C). Meanwhile, we further evaluated the effects of p38 siRNA transfection and p38 inhibitor treatment on MTERF3-mediated cell proliferation inhibition and apoptosis in LM3 cells, the results verified that inhibition of p38 activation effectively rescued the cell proliferation inhibition and apoptosis induced by MTERF3 knockdown in LM3 cells (Figure 5B-G and S7C-D). In addition, ROS scavenger Trolox treatment also significantly reduced the ROS accumulation (Figure

6G-H), and then significantly reversed the p38 signaling activation and cell proliferation inhibition induced by MTERF3 knockdown in LM3 cells (Figure 6I-J). We added these results in revised manuscript.

(7) Showing individual values is required to display variation in experiments.

Response: As the reviewer suggested, we reprepared all figures and displayed them in the revised manuscript.

Minor comments:

(1) Fig. 2J: Labeling should be improved and match the Figure legend.

Response: We are regretted that we made a mistake in labeling the Figure 2J, and we had corrected it in revised manuscript.

(2) The title of Fig. 5 should be improved.

Response: As the reviewer pointed out, we improved the title in our revised manuscript.

Referee #2:

Major concerns:

1. Most of the experiments in this manuscript depends on siRNA or shRNA knockdown. As siRNA has strong off-target effect, usually at least two siRNA/shRNA sequence and rescue experiment are needed to exclude off-target effect, especially for some key experiments. This is the standard for knockdown experiments. The authors provide two siRNA sequence data for proliferation and p38 signaling experiments, that is good. However, they should provide data for other key experiments, like Figure 2H-I, Figure 3A-B and Figure 6C-F.

Response: The reviewer made a good point. Indeed, as shown in our previous manuscript, we also considered the off-target effect of siRNA, and another siRNA (siMTERF3-1) was

used and also found that the siRNA could suppress the expression of MTERF3 and then inhibit cell proliferation and activate p38 MAPK pathway in HCC cells. As the reviewer suggested, we performed more experiments using siMTERF3-1. The results showed that, consistent with siMTERF3, siMTERF3-1 transfection also significantly inhibited the cell cycle progression and induced apoptosis in HCC-97H cells (Figure S4A-B and S4E-G). In addition, we also evaluated the effects of siMTERF3-1 transfection on mitochondrial function and ROS production. We found that siMTERF3-1 transfection also caused a significantly elevation of mtDNA transcription, led to a decrease of mtDNA content and ATP production, and triggered a strikingly ROS accumulation in HCC-97H cells (Figure S8A and S8C-F). In addition, we performed a rescue experiment to further eliminate the possibility of off-target. The results showed that MTERF3 OE could effectively restore the proliferative capacity inhibited by MTERF3 knockdown (Figure 2H and S2I). Collectively, these results intensively support that the phenotypes induced by MTERF3 knockdown is specific mediated by MTERF3 absence. We had added these results in our revised manuscript. In addition, we are sorry for that we could not perform the tumor inhibition *in vivo* with another siRNA because of the limitation in ethic and cage position of our institute.

2. One big gap of this manuscript is that how MTERF3 regulates ROS. The authors claim that MTERF3-mediated p38 MAPK activation depends on mitochondrial dysfunction-induced ROS accumulation and knockdown of MTERF3 decreases mitochondria DNA level and ATP generation. In Figure 6A-B, the authors show that knockdown of MTERF3 increases mitochondria gene expression, this is not consistent with Figure 6C which shows a decrease of mtDNA contents. Can the authors explain this? As mitochondria is one major organelle for ROS production, ROS leakage happens from ETC complex or mitochondrial respiration. So the authors should check

ETC protein levels by western blot or mitochondrial respiration by seahorse experiment. This will help explain the ROS accumulation induced by MTERF3 knockdown and makes this part much stronger.

Response: As one of the mitochondrial transcription terminators, we demonstrated that, consistent with in mice and *Drosophila*, MTERF3 knockdown caused a significant elevation of mitochondrial genes transcription in human HCC cells. However, mitochondria are semiautonomous organelle, which indicates that mitochondrial function and dynamics are strictly controlled by nuclear genome. We speculate that MTERF3 knockdown-induced elevation of mitochondrial genes transcription may feedback to nuclear genome, and causes a disorder in mitochondrial dynamics and mitochondrial dysfunction, including a decrease in copy number of mtDNA and ATP generation. Consistently, previous study also found that overexpression of mitochondrial transcription factor A (TFAM) could stimulate the transcription of mtDNA but led to a mildly decrease in abundance of full-length mtDNA (Doi: 10.1093/nar/gkh921). Nevertheless, we can not eliminate the possibility that MTERF3 has another functions independent on its role in mitochondrial transcription termination.

In addition, as the reviewer suggested, we examined the effects of MTERF3 knockdown on mitochondrial respiration via measuring oxygen consumption rate (OCR) by using an oxytherm Clark-type electrode. The results showed that MTERF3 knockdown strikingly inhibited the OCR in basal respiration and maximal respiration in HCC-97H cells, indicates that MTERF3 knockdown certainly induces mitochondrial dysfunction in HCC cells. We added the results and discussion in our revised manuscript.

3. Following point2, the authors detect ROS level here by DCF staining (2',7'-dichlorofluorescein diacetate (DCFH -DA)), which is a commonly used ROS dye. However, a lot of concerns have been raised about this ROS detection methods. Besides,

the DCF intensity not only reflects oxidation of this dye but also uptake of this dye into the cell. As ROS regulation by MTERF3 is an important part in this manuscript for the regulation mechanism, the authors should provide more solid experimental evidence for ROS detection, like using other ROS detection kit or genetically encoded ROS reporter. It would be good to detect mitochondria ROS by mitoSOX, as the model here is mitochondrial dysfunction leads to ROS accumulation for P38 signaling regulation.

Response: The reviewer made a good point. We examined the mitoROS using mitoSOX. The results also showed that MTERF3 knockdown significantly induced an increase of mitoROS in HCC-97H cells (Figure S8G-H). Therefore, we verified that MTERF3 knockdown could induce mitochondrial dysfunction and an increased production of mitochondrial ROS. We added these results in our revised manuscript.

Minor comments:

1. In Materials and Methods, the authors should give details of bioinformatics analysis for Figure 1A, B and H.

Response: As the reviewer suggested, we added the related information in Materials and Methods.

2. Does MTERF3 knockdown also affect the proliferation of normal cells, as it is a key regulator for mitochondrial gene transcription.

Response: As the reviewer suggested, we examined the effects of MTERF3 knockdown on the proliferation of LO2, a normal liver cells. The result showed that MTERF3 knockdown weakly inhibited the proliferation of LO2. However, MTERF3 knockdown did not induce a significant apoptosis in LO2 cells (Figure S5). We added these results in our revised manuscript.

3. Some words in the text needs to be changed to be more accurate. For example, in line 50, 'Silencing of MTERF3 induces mitochondrial dysfunction' should be 'Knockdown of MTERF3 induces mitochondrial dysfunction.'

Response: As the reviewer suggested, we had improved the description in our manuscript. We also checked through the manuscript before revision.

REVIEWERS' COMMENTS:

Reviewer #1 (Remarks to the Author):

I would like to thank the authors for addressing all of my comments in their revised manuscript. However, there is one point that still needs to be addressed :

1. I am somewhat confused about the results of the FLAG control group in Figure 2H and Figure S2I. Please check it. Also, I am curious to ascertain the extent of the disparity between endogenous and exogenous MTERF3 in terms of molecular weight in Figure S2I and Figure S3A.

Reviewer #2 (Remarks to the Author):

I feel like the authors appropriately responded to the concerns raised by the addition of some experiments. Equally importantly, they also added more context and nuance to the text. I think it's appropriate for acceptance now.

Thank you for the review of our manuscript entitled “Suppressing MTERF3 inhibits proliferation of human hepatocellular carcinoma via ROS-mediated p38 MAPK activation”. We are grateful to the reviewer for the constructive suggestions that have improved the quality of the manuscript. The point-to-point responses are presented below.

Referee #1 (Remarks to the Author):

1. I would like to thank the authors for addressing all of my comments in their revised manuscript. However, there is one point that still needs to be addressed. I am somewhat confused about the results of the FLAG control group in Figure 2H and Figure S2I. Please check it. Also, I am curious to ascertain the extent of the disparity between endogenous and exogenous MTERF3 in terms of molecular weight in Figure S2I and Figure S3A.

Response: We are regretted that we made a mistake in labeling the Figure 2H and Figure S2I. In order to perform the rescue assay, we transfected HCC-97H cells with siNC or siMTERF3-2 (a siRNA targeting 3’UTR region of *MTERF3* mRNA) plus indicated plasmids. Ectopic expression Flag-MTERF3 but not Flag could effectively restore the proliferation capacity suppressed by siMTERF3-2 transfection. We had corrected the label in our revised manuscript (also displayed below for convenience).

In addition, as the reviewer pointed out, there is significant disparity in molecular weight of endogenous and exogenous MTERF3. MTERF3 has two different transcript variants. To perform rescue and overexpression assays, we amplified the variant (1) using a cDNA library from 293T cells, and inserted it into the pCDH vector. The variant encodes a longer isoform containing 417 amino acids. However, based the results of Figure S2I and S3A, we found that the molecular weight of exogenous MTERF3 is about 56 kDa while endogenous MTERF3 is about 48 kDa, which means that variant (2) may be the dominant transcript variant in HCC cells, which encodes a shorter isoform containing 361 amino acids. Nevertheless, our results demonstrated that overexpression of the longer isoform also could inhibit the mtDNA transcription (Figure S8B) and promote cell proliferation and cell cycle progression (Figure S3 and S4C-D). Meanwhile, Ectopic expression of the isoform could rescue the cell proliferation capacity in MTERF3 knockdown cells (Figure 2H). Considering the minor difference in C-terminal of MTERF3 between the two isoforms and similar cellular function, we did not further construct a plasmid encoding shorter isoform and perform related assays.